# Efficacy and Residual Toxicity of Insecticides on *Plutella xylostella* and Their Selectivity to the Predator *Solenopsis saevissima*

**DOI:** 10.3390/insects14020098

**Published:** 2023-01-17

**Authors:** Daiane G. do Carmo, Thiago L. Costa, Paulo A. Santana Júnior, Weyder C. Santana, Alberto L. Marsaro Júnior, Poliana S. Pereira, Abraão A. Santos, Marcelo C. Picanço

**Affiliations:** 1Departamento de Fitotecnia, Universidade Federal de Viçosa, Viçosa 36570-900, Brazil; 2Departamento de Entomologia, Universidade Federal de Viçosa, Viçosa 36570-900, Brazil; 3Embrapa Trigo, Passo Fundo 99050-970, Brazil; 4West Florida Research and Education Center, Entomology and Nematology Department, University of Florida, Gainesville, FL 32565, USA

**Keywords:** chemical control, diamondback moth, resistance, *Brassica*, predators, fire ants

## Abstract

**Simple Summary:**

Insecticides that cause high mortality on pest insects while having a low impact on natural enemies are necessary for crop protection. Our study aimed to test commercial insecticide toxicity and residual activities against *Plutella xylostella* and the predator ant *Solenopsis saevissima*. Seven of the nine evaluated insecticides (bifenthrin, chlorfenapyr, chlorantraniliprole, cyantraniliprole, indoxacarb, spinetoram, and spinosad) caused mortality ≥80% of *P. xylostella*. In addition, four insecticides had a long-lasting effect in the field: chlorantraniliprole, cyantraniliprole, spinetoram, and spinosad. Chlorantraniliprole and cyantraniliprole always caused mortality <30% to *S. saevissima*. Furthermore, four days after application, spinetoram and spinosad caused lower mortality in the predator ant than the pest. Therefore, chlorantraniliprole and cyantraniliprole are highly recommended for controlling *P. xylostella* since they effectively control the pest with a low toxic effect on predator *S. saevissima*.

**Abstract:**

We evaluated the efficacy and residual toxicity of nine commercial insecticides on *Plutella xylostella* and their selectivity to the predator ant *Solenopsis saevissima* under laboratory and field conditions. First, to test the insecticides’ effectiveness and selectivity, we conducted concentration-response bioassays on both species and the mortalities were recorded 48 h after exposure. Next, rapeseed plants were sprayed following label rate recommendations in the field. Finally, insecticide-treated leaves were removed from the field up to 20 days after application and both organisms were exposed to them as in the first experiment. Our concentration-response bioassay indicated that seven insecticides caused mortality ≥80% of *P. xylostella*: bifenthrin, chlorfenapyr, chlorantraniliprole, cyantraniliprole, indoxacarb, spinetoram, and spinosad. However, only chlorantraniliprole and cyantraniliprole caused mortality ≤30% of *S. saevissima*. The residual bioassay indicated that four insecticides had a long-lasting effect, causing mortality of 100% to *P. xylostella* 20 days after application: chlorantraniliprole, cyantraniliprole, spinetoram, and spinosad. For *S. saevissima*, bifenthrin caused mortality of 100% during the evaluated period. Additionally, mortality rates below 30% occurred four days after the application of spinetoram and spinosad. Thus, chlorantraniliprole and cyantraniliprole are safe options for *P. xylostella* management since their efficacy favor *S. saevissima*.

## 1. Introduction

Chemical control is an effective, low-cost, easy-to-use strategy to manage pest insects [1,2]. However, detrimental effects on non-target organisms, such as natural enemies and pollinators, threaten this method’s environmental safety [3,4]. As a result, assays considering insecticide selectivity are needed by the scientific community, regulatory agencies, and the general public to determine sustainable strategies to integrate pest management into agroecosystems [4,5,6]. 

Insecticide selectivity is classified into physiological and ecological approaches [5]. The first refers to using insecticides that are more toxic to the pest than their natural enemies [5,6]. In contrast, the second refers to minimizing exposure to non-target organisms, which can be achieved through critical selection, timing, dosage, placement, and formulation [5]. In the best-case scenario, insecticides should lead to high pest mortality while having low effects on natural enemies.

A comprehensive characterization of insecticides’ physiological/ecological selectivity is crucial for integrated pest management programs. To address this, we used the diamondback moth *Plutella xylostella* (L.) (Lepidoptera: Plutellidae) and the fire ants *Solenopsis saevissima* F. Smith (Hymenoptera: Formicidae) as experimental models.

*Plutella xylostella* is one of the most important and destructive pests of rapeseed and other cruciferous crops worldwide [7]. The larvae attack the leaves, decreasing the plant’s photosynthetic area, growth, and productivity. Due to control failures and insecticide resistance, the cost of *P. xylostella* management is estimated at USD 4 to 5 billion annually [8]. Therefore, the extensive use of pesticides to control this pest represents a considerable risk to natural enemy populations [9].

Ants play a vital role in the biological control of Lepidoptera [10,11,12]. These insects forage on plants and the ground to capture lepidopteran larvae, causing a significant reduction in pest population sizes [11,12]. For instance, a recent study in Brazil indicates that predator ants, including *S. saevissima,* are the leading mortality factor of *P. xylostella* [11]. *Solenopsis saevissima* is highly active at night, builds their nests on the ground, and feeds on the larvae and pupae of many species of Lepidoptera [10,11].

Because insecticides are the primary control method against *P. xylostella* but can also cause adverse effects on non-target organisms, our study aimed to test the efficacy and residual toxicity of commercial insecticides on *P. xylostella* and whether any of these insecticides have a low impact on the predator *S. saevissima*. 

## 2. Materials and Methods

### 2.1. Crop Establishment

For the bioassays, rapeseed plants (cv. Hyola 433) were grown in an experimental field at the Universidade Federal de Viçosa (UFV), Minas Gerais State, Brazil (20°48′45′′ S, 42°56′15′′ W, 600 m altitude, and tropical climate). The rapeseed plants were cultivated according to the recommended agronomic practices [13].

### 2.2. Insects

*Plutella xylostella* larvae were collected from commercial crops of *Brassica,* and colonies were established at the IPM laboratory of UFV. The insects were reared in cages (45 × 45 × 45 cm) covered with organza. Each cage hosted a particular lifecycle stage (e.g., eggs, larvae, pupae, and adults). Egg-laying cage were set with approximately 100 *P. xylostella* adults, and *Brassica* leaves were used as an oviposition substrate. The adults were fed a 10% (*w/v*) honey solution. After hatching, the leaves with newly emerged larvae were transferred to subsequent cages, and the *Brassica* leaves were used to feed larvae until adult emergence. The *S. saevissima* adults were collected at the UFV campus from four active nests. The second instar larvae of *P. xylostella* and the adults of *S. saevissima* were used in the bioassays. 

### 2.3. Insecticides

The insecticides and concentrations used in the bioassays are described in Table 1. The insecticides were selected to cover different chemical groups and the concentrations applied were based on recommendations by the Brazilian Ministry of Agriculture Livestock and Supply to control *P. xylostella* [14]. The insecticides were diluted in water, and the adjuvant nonylphenol polyethylene glycol ether 125 SL (Milenia Agro Ciências S.A., Londrina, Brazil) at a dose of 5 mL L^−1^ was added.

### 2.4. Bioassays

Laboratory bioassays and field trials were performed. The laboratory bioassay was conducted to assess the insecticidal control efficacy of *P. xylostella* and the physiological selectivity of efficient pesticides for *S. saevissima*. The field trial was performed to evaluate the residual control period on *P. xylostella* and the selectivity on *S. saevissima*.

#### 2.4.1. Efficacy of insecticides against *P. xylostella*

To assess the insecticidal control efficacy of *P. xylostella*, rapeseed leaves (cv. Hyola 433) were immersed in each insecticide solution for five seconds. The controls consisted of water + adjuvant. The trials were completely randomized, with four replicates per insecticide and control treatment. Each replicate was a treated rapeseed leaf disc (diameter = 90 mm) placed in a Petri dish (90 × 20 mm) with 10 *P. xylostella* larvae (2nd instar). All the Petri dishes were maintained at 27 ± 2 °C, 75 ± 5% relative humidity and a photoperiod of 14:10 h (light: dark). The mortality was assessed 48 h after treatment exposure because it is considered the ideal assessment time for neurotoxic or respiratory modulator insecticides [15,16]. The insects that did not respond to stimuli with a paintbrush and remained static for more than ten seconds were considered dead.

#### 2.4.2. Physiological Selectivity to *S. saevissima*

The bioassay was carried out in a B.O.D. (biochemical oxygen demand) incubator at 28 ± 1 °C, 75 ± 5% relative humidity and no light. The bioassays were completely randomized, with four replicates for each treatment. The treatments consisted of the insecticides that were effective against *P. xylostella* in the first bioassay (i.e., mortality ≥ 80%) and a control. They were: (i) bifenthrin, (ii) chlorantraniliprole, (iii) chlorfenapyr, (iv) cyantraniliprole, (v) indoxacarb, (vi) spinetoram, and (vii) spinosad. Each replicate was composed of ten adults of *S. saevissima* in one Petri dish (90 × 20 mm) with a rapeseed leaf disc (diameter = 90 mm) treated with insecticide. The dish was covered with a fine screen mesh to allow ventilation. Food (a mix of honey and sugar in the proportion of 1:1 g) and water were provided for the insects. The mortality was assessed 48 h after exposure. 

#### 2.4.3. Residual Toxicity against *P. xylostella* and *S. saevissima*

The experiment was carried out at the UFV’s experimental field from August to October 2016. The rapeseed plants were spaced at 0.45 × 0.45 m, according to the recommended agronomic practices [13]. The field was divided into eight areas of 650 m^2,^ and each spaced 6 m apart. This procedure was performed to avoid contamination of the experimental plots during the spraying process. Each treatment consisted of either one of seven insecticides that caused mortality ≥ 80% for *P. xylostella* in the first bioassay: bifenthrin, chlorfenapyr, chlorantraniliprole, cyantraniliprole, indoxacarb, spinetoram, and spinosad. In the case of *S. saevissima*, only non-selective insecticides that caused mortality of this species ≥30% were used in the experiment: bifenthrin, chlorfenapyr, indoxacarb, spinetoram, and spinosad. The 30% threshold was chosen because insecticides that caused mortality below 30% are considered selective pesticides, according to the International Organization for Biological Control (IOBC) [6,17]. 

The spraying was performed using a CO_2_ pressurized backpack-sprayer with a pressure of 310 kPa, a flow rate of 1 L min^−1^, a volume of 240 L ha^−1,^ and spray tip type MGA 8002 (Manojet, Ind. Com. Prod. Agrícolas LTDA, Paraná State—PR, Brazil). The experimental plots were sprayed when the plants were 50 cm in height at the vegetative stage. All the procedures were used to avoid the insecticide drift to the other experimental plots. 

On each evaluation date (0, 2, 4, 6, 9, 11, 14, 15, 17, and 20 days after the insecticides were sprayed), leaves were collected from each treatment in the field. The leaves were cut into discs 90 mm in diameter in the laboratory and placed in Petri dishes (90 × 20 mm). Due to a shortage of leaves from the bifenthrin plot, the last evaluation of the predator was performed only up to 11 days after application. After these procedures, the bioassays for the pest and the non-target species were carried out. The bioassays were completely randomized, with four replicates for each treatment on each evaluation date, following the same procedures as in Section 2.4.1 for *P. xylostella* and Section 2.4.2 for *S. saevissima*. The mortality was corrected using the Abbott formula [18]. 

As noted in the laboratory bioassay, the period of control was considered the time after insecticide spraying when the *P. xylostella* mortality was ≥80%. However, the residual impact was the time after spraying that *S. saevissima* mortality was ≥30% [6,17].

### 2.5. Statistical analyses

For the laboratory (Section 2.4.1 and Section 2.4.2) and field (Section 2.4.3) bioassays, the mortality (%) of the insects was corrected using the Abbott formula [18]. Then, the experiment was performed again when the control mortality was ≥ 20%. 

In the laboratory experiment, we tested the effect of insecticides (independent variables, x-axis) on larval and ant mortality (dependent variable, y-axis). The data were analyzed using analysis of variance (ANOVA), and, before analysis, they were transformed (arcsine√x/100) to meet the assumptions of homogeneity of variance and residual normality. When the insecticides had a significant effect, we performed multiple comparisons of treatments using Tukey (α = 0.05). We also determined if the mortality caused by each insecticide varied between *P. xylostella* and *S. saevissima* using a *t*-test (α = 0.05).

The residual bioassay (Section 2.4.3) data were analyzed using a generalized linear mixed model with beta error distribution (link function = logit). We evaluated the mortality caused by insecticides to *P. xylostella* and *S. saevissima,* and for each insecticide, we compared the mortalities between species. In all the models, insecticides were set as fixed effects and days after application and replicates as random effects. This random structure was used to avoid pseudo-replication in the models and to account for differences in the number of days evaluated for each treatment. The models were checked regarding error distribution suitability, dispersion, outliers, and zero inflation. The significance of treatments was tested by a Wald Chi-square test (type III), and, when significant, the means were compared using multiple pairwise comparisons with least-square means (α = 0.05)

The analyses were performed in R version 4.0.4 and RStudio version 2022.02.3 [19] using the packages agricolae [20], DHARMa [21], emmeans [22], glmmTMB [23], and ggplot2 to design the graphics [24]. 

## 3. Results

### 3.1. Efficacy of Insecticides for Controlling P. xylostella and Their Toxicity to S. saevissima

The mortality of *P. xylostella* varied among treatments (ANOVA: F_8,27_ = 21.97; P < 0.001). From the nine commercial insecticides tested, only deltamethrin (21.40 ± 7.71%) and methomyl (41.89 ± 8.90%) caused mortality below 80% (Figure 1a). Thus, only seven insecticides were tested for selectivity against *S. saevissima*. 

*Solenopsis saevissima* mortality varied among the seven insecticides tested (ANOVA: F_6,21_ = 69.76; P < 0.001). Chlorantraniliprole (2.50 ± 4.33%) and cyantraniliprole (2.00 ± 3.46%) were less toxic (Figure 1b). The highest mortality (~100%) was found for bifenthrin, chlorfenapyr, and spinosad, while indoxacarb and spinetoram caused 72.50% (±14.79) and 77.50% (±12.99) mortality, respectively (Figure 1b).

Three insecticides caused higher mortality to the pest compared to the predator: chlorantraniliprole (t = 8.28; df = 6; P = 0.0001), cyantraniliprole (t = 12.58; df = 6; P < 0.0001), and spinetoram (t = 3.83; df = 6; P = 0.008) (Figure 1). On the other hand, the mortality observed in *S. saevissima* exposed to bifenthrin was higher than in *P. xylostella* (t = 5.23; df = 6; P = 0.002). No differences in mortality of *P. xylostella* and *S. saevissima* were recorded for chlorfenapyr (t = 0.65; df = 6; P = 0.53), indoxacarb (t = 0.90; df = 6; P = 0.40), and spinosad (t = 0.57; df = 6; P = 0.58). 

### 3.2. Residual Period of Control for P. xylostella and Selectivity to S. saevissima

Based on the concentration-response bioassays, the field experiment testing residual activity was conducted for seven insecticides for *P. xylostella* and five for *S. saevissima*.

For *P. xylostella*, four insecticides had a long-lasting effect, causing mortality of 100% during the evaluation period without any difference between them: chlorantraniliprole, cyantraniliprole, spinetoram, and spinosad (Wald χ^2^ = 260.90; df = 6; P < 0.0001; Figure 2). On the other hand, bifenthrin, chlorfenapyr, and indoxacarb reduced efficacy over time, exhibiting mortality ≥80% 3, 6, and 9 days after application, respectively (Figure 2). Overall, no differences between chlorfenapyr and indoxacarb were found, and bifenthrin caused the lowest mortality of *P. xylostella*. 

From five insecticides tested on *S. saevissima*, only bifenthrin caused mortality of 100% over time (Wald χ^2^ = 60.93; df = 4; P < 0.0001; Figure 2). No difference in mortality was detected between indoxacarb, spinetoram, and spinosad; compared to chlorfenapyr, they were less toxic. Overall, after four days, spinosad and indoxacarb had less effect on ant mortality (≤30%), while the time required for spinetoram and chlorfenapyr to achieve a similar level of mortality was 6 and 11 days, respectively (Figure 2). 

Bifenthrin (Wald χ^2^ = 165.90; df = 1; P < 0.0001), indoxacarb (Wald χ^2^ = 37.06; df = 1; P < 0.0001), spinetoram (Wald χ^2^ = 24.11; df = 1; P < 0.0001), and spinosad (Wald χ^2^ = 19.69; df = 1; P < 0.0001) caused higher mortality of *P. xylostella* compared with *S. saevissima*, while chlorfenapyr did not (Wald χ^2^ = 13.53; df = 1; P = 0.0002) (Figure 3). 

## 4. Discussion

Our study investigated the efficacy and selectivity of commercial insecticides in laboratory and field conditions. The laboratory experiments indicated seven (bifenthrin, chlorfenapyr, chlorantraniliprole, cyantraniliprole, indoxacarb, spinetoram, and spinosad) effective insecticides against *P. xylostella*, causing mortality ≥80% in 48 h. From these seven insecticides, only chlorantraniliprole and cyantraniliprole showed physiological selectivity to *S. saevissima* (i.e., mortality ≤ 30%). In the field trial, the efficacy of bifenthrin, chlorfenapyr, and indoxacarb decreased over time, while chlorantraniliprole, cyantraniliprole, spinetoram, and spinosad were consistently effective against *P. xylostella*. Conversely, only bifenthrin caused 100% mortality of *S. saevissima* over time (i.e., harmful according to IOBC class), while the other insecticides showed selectivity (mortality ≤ 30%) from 3 to 10 days after application. Our results provide helpful information for managing *P. xylostella* that may help mitigate the impact of chemical control on its natural enemy, *S. saevissima*. 

Deltamethrin was ineffective against *P. xylostella*. This failure has been detected in populations worldwide [25,26,27], and resistance has been primarily due to the increased activities of mixed-function oxidases and esterase detoxification enzymes [26,27]. Conversely, bifenthrin, another pyrethroid, was effective under laboratory conditions with reduced efficacy over time in the residual bioassay. Both insecticides belong to the same chemical sub-group but from different types. Bifenthrin is the type I pyrethroid that mainly affects insects’ central and peripheral nervous systems by interfering with sodium channels. Deltamethrin belongs to type II, whose primary target is the voltage-dependent sodium channel [28,29,30]. This difference may explain the results in our study and agree with the current scenario of deltamethrin control failure [26,27]. 

Methomyl was ineffective against *P. xylostella.* The resistance levels in field populations in Brazil have been estimated to be six times higher than in susceptible populations [28]. The high levels of field resistance could be due to multiple factors, such as successive insecticide applications with the same mode of action and the absence of other control methods [28]. Other countries have also noted such resistance [7] and concurred with our study.

Four insecticides (i.e., chlorantraniliprole, cyantraniliprole, spinetoram, and spinosad) presented high efficacy against *P. xyllostela* over time (20 days) after application in the field. Since these insecticides are systemic, the residual effect was expected for chlorantraniliprole and cyantraniliprole [29,30]. In addition, although spinetoram and spinosad act by contact and ingestion, they also present systemic action, which increases their residual activity, leading to long-term pest control, as noted in our study [31,32]. 

Diamides (i.e., chlorantraniliprole and cyantraniliprole) effectively controlled *P. xylostella* and had a low impact on the predator *S. saevissima*. Furthermore, due to the specific target site action in Lepidoptera (ryanodine receptors), these insecticides are hypothetically safe for non-target organisms [33,34]. Our experiment indicates lower toxicity of both insecticides on *S. saevissima* but high toxicity to *P. xylostella*. Therefore, chlorantraniliprole and cyantraniliprole should be used in integrated pest management programs. However, both insecticides belong to the diamide group. Thus, the rotation of the mode of action is crucial to reduce the pressure for resistance selection and preserve the efficacy of the insecticides available to control *P. xylostella*. 

Based on our results, the insecticides indoxacarb, spinosad, and spinetoram exhibit an insecticidal impact period of up to five days to the predatory ant *S. saevissima*. Therefore, these insecticides should not be used to conserve populations of this natural enemy. When these insecticides are necessary, they must be used, considering the principles of ecological selectivity. In this context, *S. saevissima* builds its nest on the ground [10,35]. Thus, applying these insecticides (i.e., indoxacarb, spinosad, and spinetoram) via irrigation or high-volume spraying should be avoided to reduce the risk of soaking the ant nests. In addition, these insecticides should not be sprayed during the night since this is the period when *S. saevissima* is more active and, thus, at significant risk of contamination [10,35]. Chlorfenapyr and bifenthrin had an insecticidal impact on *S. saevissima* for up to 9 and 11 days, respectively. For this reason, these two insecticides, especially bifenthrin, should only be used in situations where insecticide options are scarce. In conditions where they are required, the principles of ecological selectivity must be followed.

The results of this study allow the effective control of *P. xylostella* and the preservation of *S. saevissima* populations. Chlorantraniliprole and cyantraniliprole are safer options for pest management since they are efficient to control *P. xylostella* and selectivity to *S. saevissima*. In addition, these two pesticides exhibited the most prolonged residual period of control for *P. xylostella* (20 days), and they can be sprayed less frequently. Although bifenthrin showed high efficacy against *P. xylostella*, this insecticide must be used cautiously because of its higher toxicity and residual activity on the predatory ant *S. saevissima*. Additional assessment of potential sublethal effects on non-target organisms should be conducted to confirm conclusions about the physiological selectivity detected in our study.

## Figures and Tables

**Figure 1 insects-14-00098-f001:**
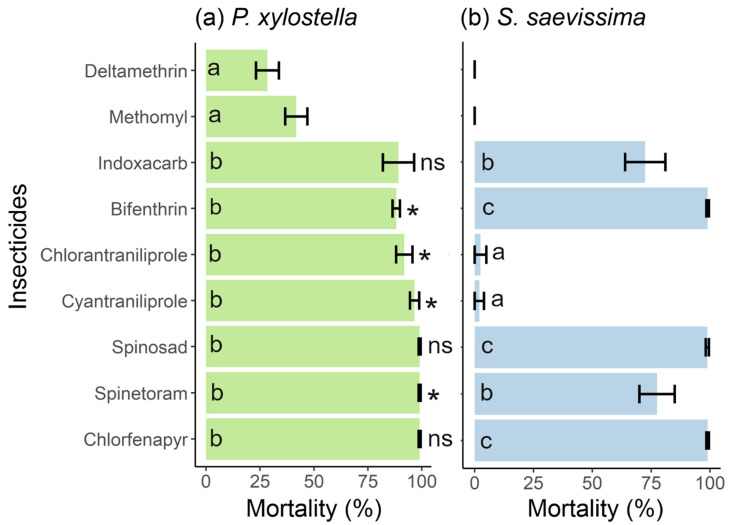
(**a**) Mortality (mean ± standard deviation) of *Plutella xylostella* larvae exposed to recommended doses of nine insecticides (ANOVA: F_8,27_ = 21.97; P < 0.001). (**b**) Mortality (mean ± error) of adults of *Solenopsis saevissima* exposed to seven insecticides that caused mortality ≥ 80% to *Plutella xylostella* (ANOVA: F_6,21_ = 69.76; P < 0.001). Different letters indicate significance among insecticides for each organism based on the Tukey test (P < 0.05). * denotes differences between the mortalities of *P. xylostella* and *S. saevissima* for the same insecticide according to the *t*-test (P < 0.05). Ns: non significative. Deltamethrin and methomyl were not tested against *S. saevissima* due to low efficacy against *P. xylostella* (mortality < 80%).

**Figure 2 insects-14-00098-f002:**
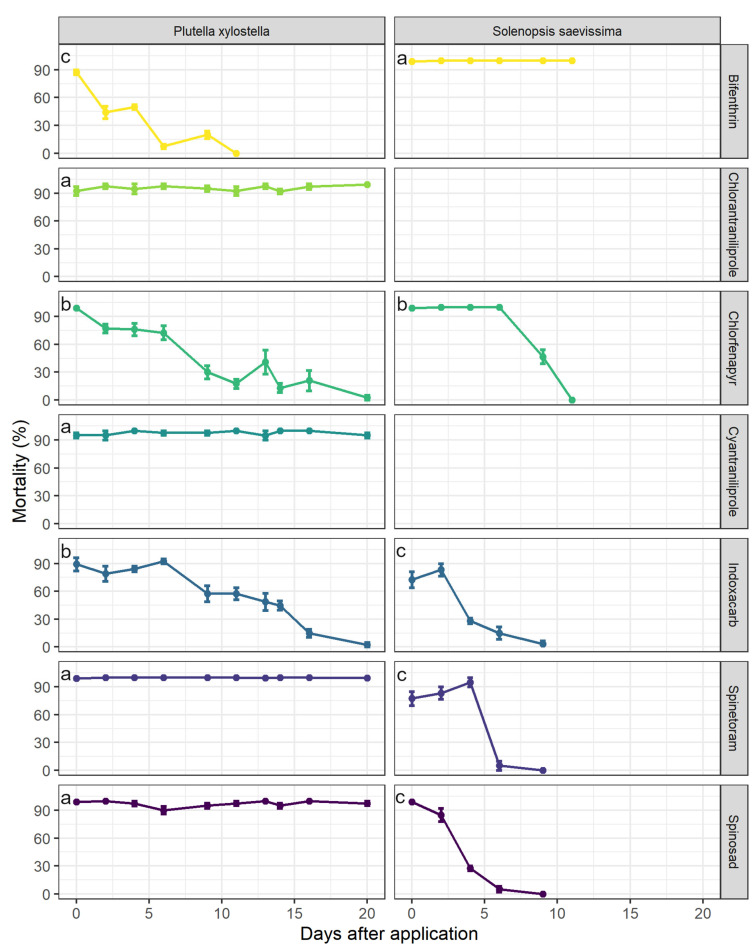
Mortality (mean ± standard deviation) of *Plutella xylostella* (Wald χ2 = 260.90; df = 6; P < 0.0001) and *Solenopsis saevissima* (Wald χ2 = 60.93; df = 4; P < 0.0001) exposed to treated leaves after the application of seven and five insecticides, respectively. Different letters indicate significance among insecticides for each organism based on least-square means at α = 0.05. Chlorantraniliprole and cyantraniliprole were not tested against *S. saevissima* because they caused lower mortality in the laboratory bioassay (see Figure 1).

**Figure 3 insects-14-00098-f003:**
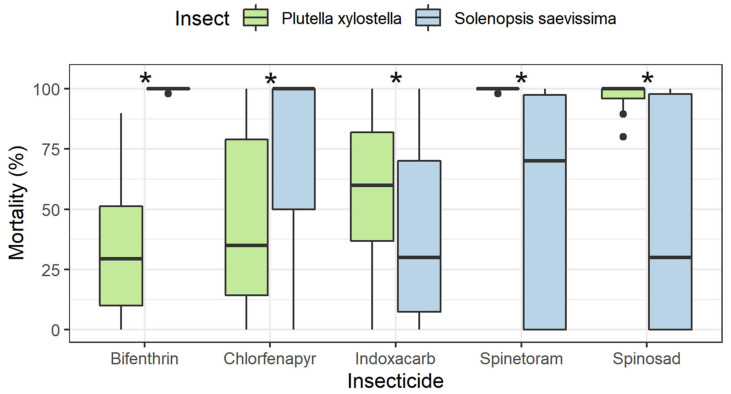
Summary of the mortality of *Plutella xylostella* and *Solenopsis saevissima* exposed to treated leaves after applying five insecticides (see Figure 2). * denotes differences between the mortalities of *P. xylostella* and *S. saevissima*: bifenthrin (Wald χ2 = 165.90; df = 1; P < 0.0001), chlorfenapyr (Wald χ2 = 13.53; df = 1; P = 0.0002), indoxacarb (Wald χ2 = 37.06; df = 1; P < 0.0001), spinetoram (Wald χ2 = 24.11; df = 1; P < 0.0001), and spinosad (Wald χ2 = 19.69; df = 1; P < 0.0001). Box plots indicate the range of data dispersion (first and third quartiles and extreme values) and median (solid line).

**Table 1 insects-14-00098-t001:** Insecticide commercial name, chemical group, manufacturer, and application rate of nine insecticides used in the study.

Insecticide	Chemical Sub-Group	Manufacturer	Rate
Bifenthrin 100 CE	Pyrethroids	FMC Química do Brasil	0.0500
Cyantraniliprole 100 OD	Diamides	FMC Química do Brasil	0.0125
Chlorantraniliprole 100 SC	Diamides	FMC Química do Brasil	0.0150
Chlorfenapyr 240 SC	Pyrroles	BASF SA	0.2400
Deltamethrin 25 SC	Pyrethroids	Bayer SA	0.0075
Indoxacarb 300 WG	Oxadiazines	Du Pont do Brasil SA	0.0300
Methomyl 215 SL	Carbamates	Du Pont do Brasil SA	0.2150
Spinosad 480 SC	Spinosyns	Corteva	0.0768
Spinetoram 120 SC	Spinosyns	Corteva	0.1200

Rate: mg a.i. mL^−1^.

## Data Availability

The datasets used and/or analyzed during the current study are available from the corresponding author (DGdC) upon reasonable request.

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
