# Peer review of "Efficacy and Residual Toxicity of Insecticides on Plutella xylostella and Their Selectivity to the Predator Solenopsis saevissima"

_insects, 2023, doi:10.3390/insects14020098_

Round 1

Reviewer 1 Report

The manuscript evaluates the laboratory mortality and field residual effect of recommended doses of several insecticides, on diamondback moth and one species of ant predator in rapeseed crop. The statistical analysis of the laboratory experiments is well performed, but the statistical analysis of the field experiment requires improvement to support the conclusions of the study. The manuscript is clearly written and the results are interesting for the integrated management of this pest. There are a few minor observations that could improve the presentation of results.

1) The statistical analysis of the field experiment needs to be improved. The IOBC criteria are good for the description and discussion of results, but they are not a statistical test. Calculation of linear regressions of insecticide mortality on time or the use of IOBC criteria, are not appropriate to conclude that the residual effects last for a particular number of days. First, it was not possible to fit the regression in several cases. Second, a repeated measures ANOVA and their multiple comparisons for each date are necessary to support this sort of conclusions. If ANOVA assumptions are not met after data transformation, then a Generalized Linear Model can be used, or a non-parametric statistical test. The authors need to provide significant differences by using letters in the Figures 3 and 4, rather than fitting a regression line and applying the IOBC criteria to conclude about residual effects.

Other minor comments

Line 55. Please use more formal language. “Therefore” instead of “So”.

Line 106. Table 1 columns about water solubility and vapor pressure of the insecticides were not used in the discussion or any other section of the manuscript. I suggest to remove this information. Manufacturer names of the companies that produce the insecticides are not updated. Cyantraniliprole, chlorantraniliprole and indoxacarb are now from FMC. Dow AgroSciences is now Corteva.

Lines 114, 136 and 153: Subsection titles 2.4.1, 2.4.2 and 2.4.3 are not in italics, but scientific names should be in italics.

Lines 125, 146: Please indicate if a maximum mortality of the control treatment was considered to include the bioassay in the study and calculate Abbot´s mortality.

Line 137: Please explain the acronym B.O.D. for the incubator.

Line 172: Weather conditions during the field trial in Table 2 were not used in the manuscript. Please include this information as supplementary material or remove.

Line 204: Figure 1. Please correct “Methomyl” instead of “Methomil”

Lines 259-260. Please clarify that both pyrethroids did not control diamondback moth in the field, but Bifenthrin showed good results in the laboratory experiment. How could this be explained?. In this paragraph please mention the possible insecticide resistance mechanism that could be involved (kdr mutation, metabolic resistance, etc.). Possible resistance to Methomyl was not discussed, please include references form Brazilian studies on diamondback moth and possible mechanisms of resistance to this insecticide as well.

Author Response

#Reviewer 1

The manuscript evaluates the laboratory mortality and field residual effect of recommended doses of several insecticides, on diamondback moth and one species of ant predator in rapeseed crop. The statistical analysis of the laboratory experiments is well performed, but the statistical analysis of the field experiment requires improvement to support the conclusions of the study. The manuscript is clearly written and the results are interesting for the integrated management of this pest. There are a few minor observations that could improve the presentation of results.

>>We appreciate your time reviewing our manuscript. The comments and suggestions helped us to provide a strong data presentation and the manuscript’s main message.

1) The statistical analysis of the field experiment needs to be improved. The IOBC criteria are good for the description and discussion of results, but they are not a statistical test. Calculation of linear regressions of insecticide mortality on time or the use of IOBC criteria, are not appropriate to conclude that the residual effects last for a particular number of days. First, it was not possible to fit the regression in several cases. Second, a repeated measures ANOVA and their multiple comparisons for each date are necessary to support this sort of conclusions. If ANOVA assumptions are not met after data transformation, then a Generalized Linear Model can be used, or a non-parametric statistical test. The authors need to provide significant differences by using letters in the Figures 3 and 4, rather than fitting a regression line and applying the IOBC criteria to conclude about residual effects.

>>Following up on the reviewer’s suggestion, we made the alterations in the analysis and data presentation.

For the laboratory data, we added the t-test to compare mortalities between organisms for the same insecticide. Please see lines 183-184 and figure 1 in line 237.

Figures 3 and 4 were removed for the residual bioassay, and the raw data is presented in figure 2 (line 248). To compare the mortality between the organisms, we created figure 3 (line 254). We performed a generalized linear mixed model with beta error distribution to compare mortalities among insecticides for the same insect and between them for the same insecticides. Although the analysis did not change the results, it provided better evidence and support the manuscript’s conclusions. Please see lines 174-197 for the full description of the statistical analysis and figures 2 and 3 in lines 248 and 254, respectively.

Other minor comments

Line 55. Please use more formal language. “Therefore” instead of “So”.

>>Corrections were made as suggested. Please see line 58.

Line 106. Table 1 columns about water solubility and vapor pressure of the insecticides were not used in the discussion or any other section of the manuscript. I suggest to remove this information. Manufacturer names of the companies that produce the insecticides are not updated. Cyantraniliprole, chlorantraniliprole and indoxacarb are now from FMC. Dow AgroSciences is now Corteva.

>>The information was updated in table 2, and table 1 was removed following reviewer’s suggestion. Please see lines 108-111.

Lines 114, 136 and 153: Subsection titles 2.4.1, 2.4.2 and 2.4.3 are not in italics, but scientific names should be in italics.

>>Corrections were made in lines 117, 128, and 139.

Lines 125, 146: Please indicate if a maximum mortality of the control treatment was considered to include the bioassay in the study and calculate Abbot´s mortality.

>>The maximum mortality allowed in control was up to 20%. We have indicated in lines 176-177.

Line 137: Please explain the acronym B.O.D. for the incubator.

>> This information is included now. Please see line 129.

Line 172: Weather conditions during the field trial in Table 2 were not used in the manuscript. Please include this information as supplementary material or remove.

>> We have removed the table.

Line 204: Figure 1. Please correct “Methomyl” instead of “Methomil”

>> The name was correct to Methomyl. Please see figure 1, line 237.  

Lines 259-260. Please clarify that both pyrethroids did not control diamondback moth in the field, but Bifenthrin showed good results in the laboratory experiment. How could this be explained?. In this paragraph please mention the possible insecticide resistance mechanism that could be involved (kdr mutation, metabolic resistance, etc.). Possible resistance to Methomyl was not discussed, please include references form Brazilian studies on diamondback moth and possible mechanisms of resistance to this insecticide as well.

>> We have included a possible explanation for the results regarding the pyrethroids in lines 275-284. The results could be associated with the subgroup of the insecticides, as bifenthrin belongs to subgroup I of pyrethroids (they lack a cyan moiety in the alpha position), while deltamethrin belongs to subgroup II (they have an alpha-cyan moiety). [1–3].

There is no study regarding potential molecular mechanisms associated with methomyl's possible resistance in P. xylostella populations in Brazil. Most studies investigated the resistance ratio between field populations from areas with higher or lower insecticide applications. We have included this information in the discussion (lines 286-290). Then we decided not to include speculative statements about it. In addition, we highlight that our manuscript did not aim to verify or determine resistance mechanisms associated with the insecticides tested. Instead, we focused on determining their efficiency on the pest and selectivity to natural enemies. Please see lines 80-83.

  1. Onstad, D.W. (Ed. Insect Resistance Management: Biology, Economics, and Prediction.; Academic Press., 2013;
  2. Schleier, J.J.; Peterson, R.K.D. The Joint Toxicity of Type I, II, and Nonester Pyrethroid Insecticides. J. Econ. Entomol. 2012, 105, 85–91, doi:10.1603/EC11267.
  3. Sparks, T.C.; Crossthwaite, A.J.; Nauen, R.; Banba, S.; Cordova, D.; Earley, F.; Ebbinghaus-Kintscher, U.; Fujioka, S.; Hirao, A.; Karmon, D.; et al. Insecticides, Biologics and Nematicides: Updates to IRAC’s Mode of Action Classification - a Tool for Resistance Management. Pestic. Biochem. Physiol. 2020, 167, 104587, doi:10.1016/j.pestbp.2020.104587.

Reviewer 2 Report

This is a very fine paper, which adressess the issue of how to combine chemical and biological pest control so that the toxicity of pesticides is high for the insect that is damaging the crop and simultaneously as little toxic as possible for the insect predator that attacks the pest insect. Moreover, the paper masterfully combines lab experiments with field experiments, and also integrates the ecology of the predator in the picture so that, when a pesticide is applied that also is harmful to the predator, it is applied in such a way that it minimizes mortality of the predator. All in all, I find that this paper is very fine and interesting, and in my opinion it can be published as it is.

Author Response

#Reviewer 2

This is a very fine paper, which adressess the issue of how to combine chemical and biological pest control so that the toxicity of pesticides is high for the insect that is damaging the crop and simultaneously as little toxic as possible for the insect predator that attacks the pest insect. Moreover, the paper masterfully combines lab experiments with field experiments, and also integrates the ecology of the predator in the picture so that, when a pesticide is applied that also is harmful to the predator, it is applied in such a way that it minimizes mortality of the predator. All in all, I find that this paper is very fine and interesting, and in my opinion it can be published as it is.

>>We appreciate your time reviewing our manuscript. We have made some improvements based on the reviewer’s 1 and 3 raisings and the manuscript’s main message is clearly presented in the new version.

Reviewer 3 Report

See attached

Author Response

#Reviewer 3

The authors have addressed the pesticide trials against DBM and their effect on its ant predator under lab and field conditions. The pest is serious in nature and inflicts heavy losses to many crops, thus the study is important. The write-up is largely OK, but still needs a bit of improvement in some places in terms of English or rephrasing. The data could be just enough to be published, although more can be done further. I, however, have a few concerns that need to be addressed before it gets published.

>> We appreciate your time reviewing our manuscript and indicating unclear statements in the first version. Based on that, we have amended it to present a clear message of the manuscript. Below, we provided the answers to your comments and questions and indicated where alterations were made in the manuscript. Thank you.

Major

  1. Any possible explanation as to why bifenthrin works and deltamethrin doesn’t against Plutella? Both are Pyrethroids, with little or no differences in mode of action if used in the same formulation.

>> The difference could be associated with the subgroup of each insecticide. Bifenthrin belongs to type I pyrethroids that mainly affect insects’ central and peripheral nervous systems by interfering with sodium channels. Deltamethrin belongs to type II, whose primary target is the voltage-dependent sodium channel. This information has been included in the manuscript. Please see lines 275-284

  1. If bifenthrin is giving you more than 80% mortality after 48h in lab and for 3 days in field, then the statement that bifenthrin doesn’t work efficiently for Plutella doesn’t seem right, as you have stated in your discussion (Lines 259-260). If it is causing more than 80% mortality, then calling it a resistance development may also be not correct. Three days duration I reckon is enough for a pesticide to work in the field. The increased persistence in the field on the other hand may also be looked at as environmental hazard, particularly in edible plants. The suggestion thus here is to rewrite this part of discussion.

>> We agree with the reviewer’s discussion regarding the misuse of the lower efficiency of bifenthrin, and we made corrections in lines 275-284.

Evaluating insecticide’s efficiency under field conditions leads to such discussion regarding its residual effect. From the farmer’s view, it is desirable that insecticides have a long last effect to avoid repeated applications to achieve control. This case seems applicable in the P. xylostella scenario, as resistance has been involved in the repeated application of the same insecticides in the fields (lines 288-289). However, on the other side, this long-lasting effect may harm and cause non-desirable effects on non-target organisms, as the reviewer states.

In our work, we demonstrated that even bifenthrin controls P. xylostella under laboratory conditions and had a short residual on P. xylostella; it caused higher mortality to the natural enemy even when it was not more efficient for the pest (Figures 2).

  1. Line 303 - Again contradictory to say that “bifenthrin showed high efficiency”, whereas elsewhere stated that bifenthrin doesn’t work. Correction needed.

>>We have made corrections in lines 331-333.

  1. The figures are a bit confusing and you have to switch over from the pest mortality figure to that of the predator to compare. It will be better to combine at least the first two figures, i.e., to plot Plutella mortality as bars and the Solenopsis mortality as a line graph over it for easy comparison of efficiency and safety of these insecticides. I reckon 9 and 7 treatments won’t matter much in terms of plotting together.

>>We welcome that suggestion and modified data presentation. Figures 1-3 have been altered, including in the same plot P. xylostella and S. saevissima. Please see lines 237, 248, and 254.  

  1. The present bioassays were conducted under no choice test. Any Choice test to consider (in future) will be a good idea for Plutella or Solenopsis?

>> The experiment would be helpful to understand that, but it was not possible to perform during the time. We included in the first version a consideration regarding the sublethal effects on both species in lines 322-326.

  1. Finally, why only Solenopsis was considered? There are plenty of natural enemies, including parasitoids. of Plutella to be considered. Is Solenopsis that much effective when compared to other natural enemies?

>> Solenopsis saevissima was chosen because this species has been indicated, together with spiders, as the leading mortality factor of P. xylostella in the region where our experiment was performed. Please see lines 72-79 and the reference:

Farias et al., 2021. Life tables for the diamondback moth (Plutella xylostella) in southeast Brazil indicate ants and spiders as leading mortality factors. Annals of Applied Biology. DOI: https://doi.org/10.1111/aab.12656

Minor

Line 46. Not only cast some doubt but for sure chemicals are threat to environmental safety.

>>Minor corrections were made to this statement. Please see lines 49-50.

Lines 158-160: There is some confusion as the authors stated “Each area received only one treatment that consisted of seven insecticides that caused mortality ≥ 80% for P. xylostella in the first bioassay: bifenthrin, chlorfenapyr, chlorantraniliprole, cyantraniliprole, indoxacarb, spinetoram and spinosad” giving the impression that seven insecticides were used in each plot. I think the authors meant to say that each treatment consisted of either of the seven insecticides. This needs to be corrected.

>>The statement was corrected in lines 144-147.

Line 161-162: Again not very clear to me as written “In the case of S. saevissima, only those insecticides that caused mortality ≥ 30% were used in the experiment:” Does it mean mortality to S. saevissima or Plutella? If it means mortality to the predator, then I reckon it should be less than 30% not more than 30%

>> We indicated that the threshold refers to S. saevissima in lines 147-149. In this bioassay, we aimed to verify how long a non-selective insecticide (based on the laboratory bioassay) caused mortality to this predator.

Lines 282-283: “To preserve the populations of this natural enemy, these insecticides should be used considering the principles of ecological selectivity”. I think you should write instead “should not be used”.

>> Thank you for that suggestion. It was altered in lines 309-311.